# Versatile Mode-Locked Operations in an Er-Doped Fiber Laser with a Film-Type Indium Tin Oxide Saturable Absorber

**DOI:** 10.3390/nano9050701

**Published:** 2019-05-05

**Authors:** Quanxin Guo, Jie Pan, Dengwang Li, Yiming Shen, Xile Han, Jinjuan Gao, Baoyuan Man, Huanian Zhang, Shouzhen Jiang

**Affiliations:** 1Collaborative Innovation Center of Light Manipulations and Applications in Universities of Shandong, Shandong Normal University, Jinan 250014, China; 2017020516@stu.sdnu.edu.cn (Q.G.); sdnupanjie@163.com (J.P.); dengwang@sdnu.edu.cn (D.L.); tsuna2013@outlook.com (Y.S.); hxlsdnu@163.com (X.H.); jinjuan_gao@163.com (J.G.); byman@sdnu.edu.cn (B.M.); 2Shandong Key Laboratory of Medical Physics and Image Processing and Shandong Provincial Engineering and Technical Center of Light Manipulations, School of Physics and Electronics, Shandong Normal University, Jinan 250358, China; 3Shandong Provincial Key Laboratory of Optics and Photonic Device, Jinan 250014, China

**Keywords:** indium tin oxide, nonlinear absorption properties, mode-locked fiber laser, saturable absorber, Er-doped fiber laser

## Abstract

We demonstrate the generation of versatile mode-locked operations in an Er-doped fiber laser with an indium tin oxide (ITO) saturable absorber (SA). As an epsilon-near-zero material, ITO has been only used to fashion a mode-locked fiber laser as an ITO nanoparticle-polyvinyl alcohol SA. However, this type of SA cannot work at high power or ensure that the SA materials can be transmitted by the light. Thus, we covered the end face of a fiber with a uniform ITO film using the radio frequency magnetron sputtering technology to fabricate a novel ITO SA. Using this new type of SA, single-wavelength pulses, dual-wavelength pulses, and triple-wavelength multi-pulses were achieved easily. The pulse durations of these mode-locked operations were 1.67, 6.91, and 1 ns, respectively. At the dual-wavelength mode-locked state, the fiber laser could achieve an output power of 2.91 mW and a pulse energy of 1.48 nJ. This study reveals that such a proposed film-type ITO SA has excellent nonlinear absorption properties, which can promote the application of ITO film for ultrafast photonics.

## 1. Introduction

Mode-locked fiber lasers are vital for modern research and applications, e.g., for optical communication, radar systems, and material processing [1,2,3]. A saturable absorber (SA) is the most important device in a mode-locked fiber laser, which can emit ultra-short pulses. Various soliton phenomena could be obtained using SAs fabricated by diverse materials. Additionally, multi-wavelength fiber lasers have attracted much research interest for wide application in ultrafast optics, optical communications, biomedical research, and radar systems. Using the SAs with high nonlinearity in the fiber laser cavity, the multi-wavelength pulses could be achieved [4,5,6,7]. Accordingly, the fabrication of novel and high performance SAs from various two-dimensional (2D) materials have become a popular field in ultrafast laser research.

Recently, 2D materials have been investigated as SAs for fashioning pulsed fiber lasers because of the outstanding saturable absorption properties [8]. Single-wall carbon nanotube (SWCNT) is an ideal material for SAs due to its various properties like broadband saturable absorption, ultrafast recovery time, and so on. Set et al. reported on passively mode-locked fiber lasers with SWCNTs as mode lockers in 2004 [9]. Various mode-locked lasers incorporating SWCNTs as SAs have been studied since then [6,10,11]. However, the strong optical scattering limited the application of SWCNTs-based SAs [12]. Graphene was successfully applied as SAs in pulsed lasers for its ultra-broad absorption band [13,14,15,16,17,18,19]. However, the weak modulation depth caused by the weak absorption efficiency limits the further application of graphene in pulsed fiber lasers [19]. Recently, transition metal dichalcogenides (TMDs) [7,20,21,22,23,24,25,26], topological insulators (TIs) [27,28,29,30,31,32,33,34,35], black phosphorus (BP) [36,37,38,39], and other materials [40,41,42,43] have also extensively served as SAs in pulsed fiber lasers. In 2013, Wang et al. achieved a short relaxation time (~30 fs) and a high saturable absorption using MoS_2_, which suggests that this material can be a high-performance SA in ultrafast lasers [44]. However, the optical bandgap limited the application of TMDs devices in the mid-infrared (MIR) region [45]. BP, an increasingly popular 2D material, has aroused the attention of researcher. Compared with TMDs, BP can serve as an excellent SA in near-infrared (NIR) and MIR regions for the large saturable absorption bandwidth. However, the weak thermal stability has limited its developments in fiber lasers [46]. To sum up, these 2D materials have great significance for the progress of mode-locked fiber lasers. In the meantime, the research of high-performance SAs with large saturable absorption, high damage threshold, ultra-fast recovery time, and broadband absorption will continue unabated.

In the past few years, ITO have been extensively studied as transparent electrodes, solar cells, and so on. In 2016, Alam et al. demonstrated the ultrafast and large intensity dependent refractive index of ITO in the epsilon-near-zero (ENZ) region [47]. In the ENZ region, ITO nanoparticles (NPs) exhibit a fast response time (~360 fs) and a deep modulation depth up to circa 160%. Because of the large third order nonlinearity and tunable optical loss, ITO can be used to fabricate switched devices [48]. These strong optical nonlinearities allow ITO to serve as ultrafast photonics devices. Besides, the absorption peak could be regulated from 1600 to 2200 nm so that the working range of the devices fabricated by ITO is relatively wide [49]. Furthermore, ITO has a high damage threshold, which allows it to work at a high pump power for a long time [47,50]. Several methods have been used to fabricate the ITO SAs, e.g., adding ITO NPs to a polyvinyl alcohol (PVA) or poly (methyl methacrylate) (PMMA) host [48,51] and dripping the ITO dispersion solution on the end face of a fiber directly [52]. However, the composite films SA cannot work at a high power due to the weak thermal capacity of PVA and PMMA. In addition, the light absorption and reflection of the polymer matrix would cause an insertion loss in the laser cavity. Moreover, these methods should ensure that the SA materials are transmitted by the light. Accordingly, a new type of SA with excellent thermal stability, low insertion loss, large scale, and extraordinary continuity is eagerly anticipated.

In this study, we report on a new type of ITO SA fabricated by depositing a uniform ITO film on a fiber head by radio frequency (rf) magnetron sputtering. Magnetron sputtering is a common method for depositing films on a substrate. The film fabricated using this method had a large scale, and its thickness can be controlled by regulating the sputtering time. Moreover, without the assistance of PVA, which could bring a large insertion loss and a low damage threshold, the ITO film exhibits a much lower insertion loss of 1.4 dB and a better thermal stability than the ITO-PVA SAs. The saturable absorption properties of the ITO SA were tested. The modulation depth and saturation intensity were nearly 8.3% and 11.9 MW/cm^2^, respectively. Using the novel ITO SA for an Er-doped fiber laser (EDFL), single-, dual-, and triple-wavelength pulses were achieved by regulating the polarization controller (PC) and pump power. The experimental results fully proved that the ITO film fabricated by rf magnetron sputtering could be a useful device for ultrafast photonics.

## 2. Materials and Methods

### 2.1. Preparation of ITO-Based SA

The ITO films were grown on the end face of a fiber by rf magnetron sputtering using a ceramic ITO target with a high purity of circa 99.99%. The atomic ratio of In:Sn in the ITO target was 90:10. The fiber was vertically fixed on a stable pedestal to ensure that the ITO film could be uniformly deposited on its end face. The distance from substrate to target was 90 mm. The chamber was pumped to 2 × 10^−5^ Torr, and the automatic sputtering process was performed in Ar (99.99%) at a flow rate of 30 sccm. After 5 min of sputtering at a rf sputtering power of 100 W, the film-type ITO SA was prepared.

### 2.2. Characterization

After the ITO SA was prepared, the scanning electron microscope (SEM) (Sigma 500, ZEISS, Oberkochen, Germany) image of the end face of fiber head was investigated, as shown in Figure 1a. The red circle shows the position of the fiber core. The SEM image of the ITO film at the end face of the fiber core is shown as the inset in the Figure 1a. The film consisted of ITO NPs about 15 nm in size. Obviously, the whole end face was covered with a smooth and large-scale ITO film, so the light could be transmitted though the SA material. To characterize the roughness and the thickness of the ITO film, an atomic force microscope (AFM) (Multimode 8, Bruker, Germany) was employed. The ITO film was deposited on a SiO_2_ substrate under the same conditions. The end face of the fiber core that was used in this experiment was made of SiO_2_, so the parameters (including the thickness, the roughness, and so on) of the ITO film deposited on the SiO_2_ substrate should be the same as the ITO film deposited on the fiber core. According to the 3-dimentional (3D) AFM image, as shown in Figure 1b, the roughness of the ITO film was Ra 0.73, suggesting that our ITO film was quite smooth. The AFM image is shown in Figure 1c, and the corresponding height measurement is displayed in Figure 1d. The thickness of the ITO film investigated was 60 nm. By controlling the sputtering time, the thickness can be changed easily for different requirements.

The crystal structure of ITO film was tested using an X-Ray diffraction (XRD)(D8 Advance, Bruker, Billerica, MA, USA). Some diffraction peaks corresponding to the (211), (222), (400), (411), (440), (611), and (622) planes in ITO are clearly observed in Figure 2a, respectively, which indicate that the ITO film is of high-quality. To analysis the elemental composition of our sample, the energy dispersive spectrometer (EDS) (QUANTAX EDS, Bruker, Germany) spectrum of the ITO film was investigated, as shown in Figure 2b. The peaks attributed to In and Sn are observed. Moreover, the atomic ratio of In to Sn, which is 91.83:8.17, is shown inset in Figure 2b. It is noteworthy that compared with the target, the atomic ratio of ITO film is slightly misaligned. This may because that a few Sn atoms were not smoothly deposited on the SiO_2_ during the deposition process. As a result, the position of the absorption peak may have a red shift compared to that from the ITO film with an In/Sn molar ratio of 90:10 [53].

The linear absorption spectrum of ITO film was measured with a UV/vis/NIR spectrophotometer (U-4100, Hitachi, Tokyo, Japan), as shown in Figure 3a. The detection light sources were a xenon lamp (for the UV region) and a tungsten halogen lamp (for the vis/NIR region) with a mW-level-power. Obviously, the absorbance increased with the wavelength. At 1558 nm, the absorbance was measured as 4.8%. The absorption bandwidth of ITO film was quite wide in the NIR, allowing for its application of photonics devices in the NIR. Additionally, the nonlinear optical absorption was investigated using a typical balanced twin-detector method. The pump source used here was a home-made nonlinear polarization rotation mode-locked EDFL with 560 fs pulses at the wavelength of 1560 nm and a repetition rate of 33.6 MHz. The recorded results are shown in Figure 3b. The transmittance shows a nonlinear dependence on the pump power, and the saturable absorption property of the ITO film can be obtained by analyzing this relation. By fitting the experimental results, the modulation depth and saturation intensity were calculated, which were 8.3% and 11.9 MW/cm^2^, respectively.

To highlight the advantages of the proposed ITO film SA prepared by magnetron sputtering deposition technique, we compared it with the PVA-type SA here. Firstly, the light absorption and reflection of PVA would cause an insertion loss in the laser cavity. Using an EDFL at a wavelength of 1560 nm, the insertion loss was experimentally recorded to be 1.4 dB. Compared to the PVA-type SAs (the insertion losses were 5.5, 2.5, 3.5, and 6.1 dB, respectively) [42,54,55,56], the insertion loss of the ITO film SA was significantly lower. Secondly, the PVA-type SA cannot work with a high power due to the weak thermal capacity of PVA. Without the limitation of PVA, the damage threshold can be significantly increased, offering the possibility to work under high-energy conditions. 

### 2.3. Mode-Locked Fiber Laser

Figure 4 shows the configuration of the mode-locked fiber laser. We used a 974 nm fiber-pigtailed laser diode (LD) (Jinan Jingjiang photoelectric technology co. LTD, Jinan, China) as the pump source. The laser cavity consists of a piece of Er-doped fiber (EDF) (Liekki Er-110, nlight, Vancouver, WA, USA) with a dispersion parameter of −46 ps/nm/km, and a piece of single-mode fiber (SMF, SMF-28) with a dispersion parameter of 18 ps/nm/km. For the EDF, the peak core absorption and the concentration of the Er^3+^ ion were 110 dB/m and 4350 ppm, respectively. The length of the EDF and SMF were 24 cm and 100 m, respectively. A 980/1550 wavelength-division multiplexer (WDM) (Jinan Jingjiang photoelectric technology co. LTD, Jinan, China) was used as the coupler, and a 90/10 optical coupler (OC) (Jinan Jingjiang photoelectric technology co. LTD, Jinan, China) was used to export the 10% laser. There were two PCs to adjust the cavity birefringence in our fiber laser. Moreover, to ensure unidirectional light propagation, a polarization independent isolator (PI-ISO) (Jinan Jingjiang photoelectric technology co. LTD, Jinan, China) was used. The total length and dispersion were 104.8 m and −2.204 ps^2^, respectively.

## 3. Results and Discussion

In this section, we discuss three mode-locked operations in our EDFL based on the ITO film SA. In our experiments, the mode-locked operation was obtained with a length of the cavity at 104.8 m. It is generally known that self-mode-locked operation can sometimes be obtained with a long-length cavity, so the fiber laser was tested without using ITO SA. By regulating the pump power and PCs carefully, no pulse operation was observed, indicating that there was no self-mode-locked operation in the cavity. Using the ITO film SA in the cavity, single-, dual-, and triple-wavelength pulses were regulating by adjusting the PCs and pump power.

### 3.1. Single-Wavelength Pulses

Mode-locked state was achieved at the pump power of 287 mW by adjusting the two PCs. The output optical spectrum with the central wavelength at 1558.5 nm is provided in Figure 5a. In addition, the 3-dB bandwidth is 0.27 nm. However, the output optical spectrum did not show the Kelly sideband. This proved that the mode-locked operation was not at the conventional soliton mode-locked state. According to the zoom-in single pulse profile, as shown in Figure 5b, the pulse duration is nearly 1.67 ns. The time bandwidth product (TBP) is about 55.7, much larger than the theorical value of 0.315. This reveals that the optical pulse was highly chirped, primarily due to the long length of the laser cavity. Regulating the length of the total cavity is a common strategy to compress the pulse width. However, the long length of the laser cavity contributes to the generation of solitons, so that the length of the cavity cannot be arbitrarily shortened. As shown in Figure 5c,d, the repetition rate was 1.96 MHz, corresponding to the total laser cavity length. The signal-to-noise ratio (SNR) is 40 dB, suggesting the mode-locked operation exhibited a good stability. Moreover, the measurement of the radio frequency (RF) spectrum in a span of 100 MHz further proved the stability of the mode-locked operation (inset in Figure 5d).

### 3.2. Dual-Wavelength Pulses

At the pump power of 268 mW, the dual-wavelength pulses were achieved easily by regulating the PCs. The output optical spectrum is shown in Figure 6a and contains two wavelengths and centered at 1557.6 and 1558.5 nm. The 3-dB bandwidths of the two wavelengths are 0.130 and 0.331 nm, respectively. A single-pulse, as shown in Figure 6b, has a pulse duration of 6.91 ns. The inset of Figure 6b shows the typical pulse train with a period of 510.2 ns, which matches the roundtrip time of the cavity and verifies the mode-locked operation. It is worth mentioning that the trace of pulses exhibits an amplitude envelope modulation. This phenomenon was due to the power instability and the phonon vibration of the SA material. The RF spectrum, which could test the stability of the mode-locked fiber laser, is shown in Figure 6c. The peak observed at the repetition rate of 1.96 MHz has a SNR of 50 dB, suggesting that the mode-locking state exhibits an excellent stability. Moreover, the RF spectrum was also measured within a wide bandwidth of 100 MHz, as shown in the inset of Figure 6c, which further verified that the mode-locked operation exhibits a high stability. As shown in Figure 7, by fixing the PCs at a dual-wavelength pulses state, the output optical spectra and single pulse versus the pump power were recorded. The central wavelength and the 3-dB bandwidth are nearly unchanged while the pump power increased, as shown in Figure 7a. Figure 7b shows the dependence of the pulse width versus the pump power. The pulse width was fixed at about 6.91 ns, and the intensities were up-regulated while the pump powers were regulated from 268 to 310 mW. The output power was raised from 1.73 to 2.91 mW as the pump power was regulated from 254 to 403 mW. According to the repetition frequency of 1.96 MHz, the single pulse energy increased from 0.89 to 1.48 nJ, as shown in Figure 6d.

### 3.3. Triple-Wavelength Multi-Pulses

Triple-wavelength multi-pulses were achieved by adjusting the PCs carefully at a higher pump power of ~420 mW. According to the analysis of the output optical spectrum, as shown in Figure 8a, the central peak of the spectrum was located at 1558.5 nm, which was the same as the main peak from the spectra of the single- and dual-wavelength pulses. The 3-dB bandwidth were all about 0.20 nm. Figure 8b shows the pulse train of the triple-wavelength multi-pulses. The period of the pulses train was 510.2 ns, which implies that the fundamental repetition rate remained 1.96 MHz. The zoomed-in view of a single pulse with a pulse width of about 1 ns, which has a symmetric hyperbolic secant intensity profile, is shown in Figure 8c.

In the experiments conducted in this study, it is noteworthy that the multi-wavelength mode-locked operations were obtained without comb filter in our fiber laser. The polarization-dependent loss attributed to the PCs and birefringence of the SMF can cause the comb filtering effect. Furthermore, there was a large nonlinearity in the laser cavity, which was attributed to the large third-order nonlinear coefficient of ITO film. For these reasons, the multi-wavelength pulses could be achieved in our fiber laser naturally [57].

## 4. Conclusions

To sum up, the film-type ITO SA was first experimentally demonstrated using the rf magnetron sputtering technique. The film-type ITO SA fabricated using this method exhibits an excellent thermal stability, low insertion loss, large scale, and prominent continuity. Using such SA, the single-, dual-, and triple-wavelength pulses were achieved in an EDFL. Our study fully indicates this film-type ITO SA fabricated by rf magnetron sputtering exhibits excellent nonlinear absorption characteristics, which can promote the use of ITO film in the field of ultrafast photonics.

## Figures and Tables

**Figure 1 nanomaterials-09-00701-f001:**
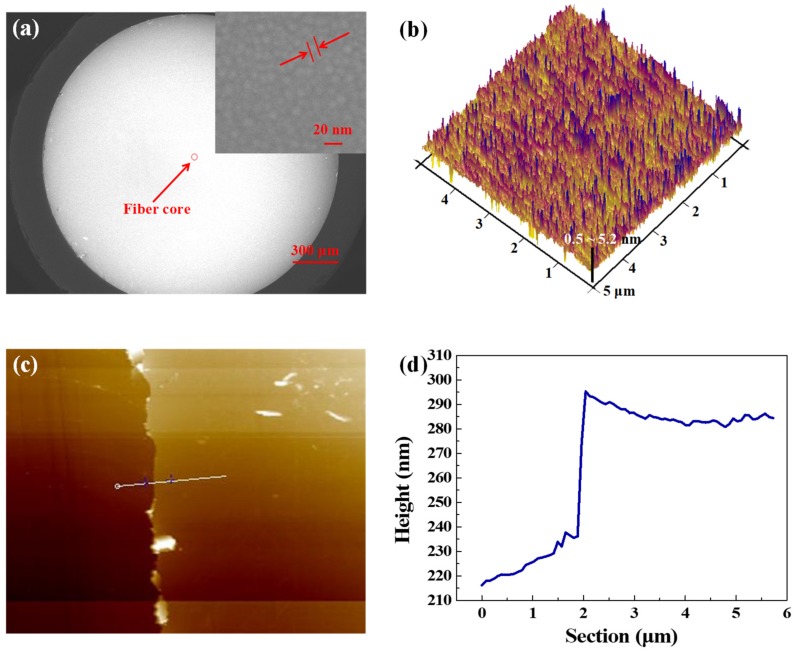
(**a**) Scanning electron microscope (SEM) image of the fiber connector after depositing indium tin oxide (ITO) film (inset, the SEM investigated at the center region), (**b**) the 3D atomic force microscope (AFM) image of the ITO film on SiO_2_ substrate, (**c**) the AFM image of the ITO film, and (**d**) the height measurement of the selected area in (**c**).

**Figure 2 nanomaterials-09-00701-f002:**
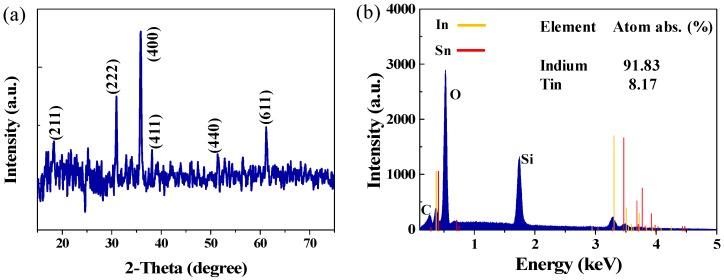
(**a**) The X-Ray diffraction (XRD) image of the ITO film, and (**b**) the energy dispersive spectrometer (EDS) of the ITO film.

**Figure 3 nanomaterials-09-00701-f003:**
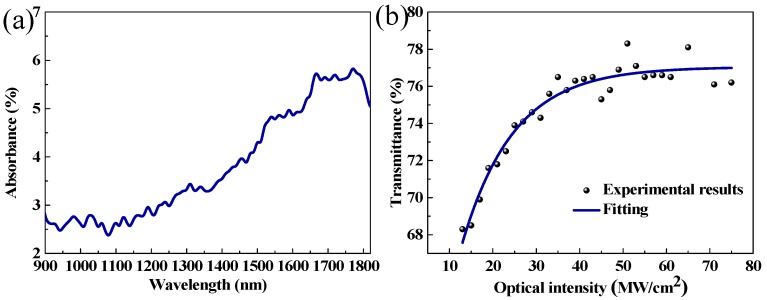
(**a**) The measured linear absorption; (**b**) the measured saturable absorption curve.

**Figure 4 nanomaterials-09-00701-f004:**
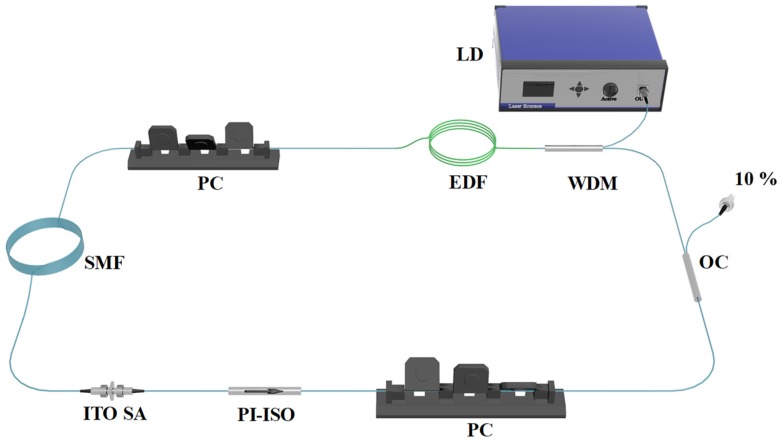
Schematic of the pulsed fiber laser including the ITO film saturable absorber (SA).

**Figure 5 nanomaterials-09-00701-f005:**
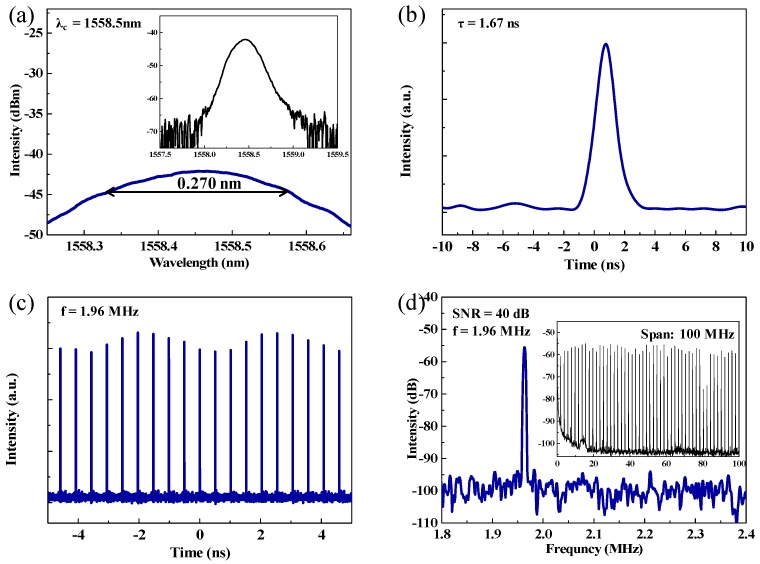
(**a**) The output optical spectrum, (**b**) a zoom in single-pulse profile, (**c**) the pulse train, and (**d**) RF spectrum (inset, spectrum measured in 100 MHz range).

**Figure 6 nanomaterials-09-00701-f006:**
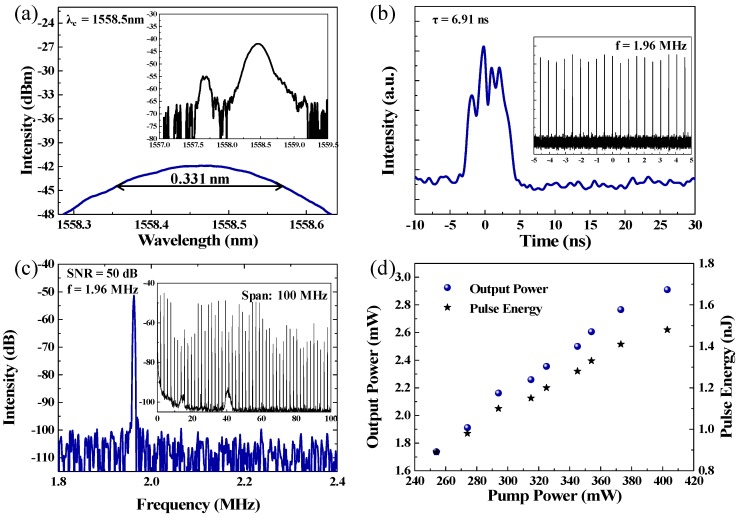
(**a**) The output optical spectrum, (**b**) a zoomed–in view of a single-pulse profile, and the inset is the pulse train, (**c**) RF spectrum (insert, RF spectrum with a bandwidth of 100 MHz range), and (**d**) output power of pulse and pulse energy versus power.

**Figure 7 nanomaterials-09-00701-f007:**
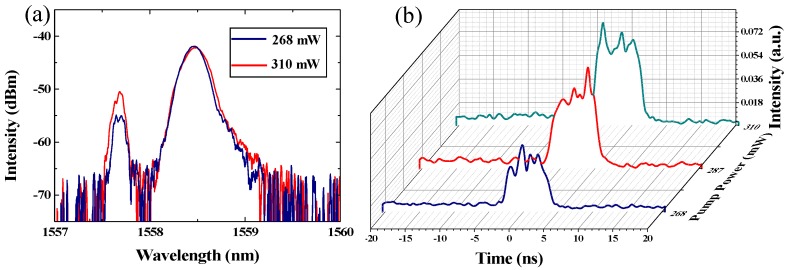
(**a**) Output optical spectra versus pump power, and (**b**) pulse waveform versus pump power.

**Figure 8 nanomaterials-09-00701-f008:**
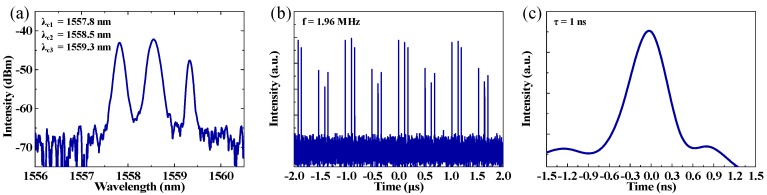
(**a**) Output optical spectrum, (**b**) typical pulse train, and (**c**) zoom in pulse profile.

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
