# Peer review of "Versatile Mode-Locked Operations in an Er-Doped Fiber Laser with a Film-Type Indium Tin Oxide Saturable Absorber"

_nanomaterials, 2019, doi:10.3390/nano9050701_

Reviewer 1 Report

This paper demonstrates a mode-locked operations in an fiber laser by using ITO SA. Compared with previously reported ITO SAs such as nanoparticles, nanoparticles in PVA or PMMA, they achieved uniform ITO film, which can be expected to endure large optical power and low insertion loss. Authors reported single-, dual-, triple-wavelength multipulses. The idea of ITO film SA is interesting, however, the detailed comparisons with previous researches and more analysis should be revised for the publication.

Authors emphasized that ITO film can have excellent thermal stability, low insertion loss, and large scale. However, in the paper, they did not reported the threshold of optical power, insertion loss of their ITO films. Authors should discuss such values and compare them with those values of the previously reported ITO SAs.

In Fig.1, in order to estimate the roughness of ITO film, authors used SiO2 substrate instead of optical fiber. However, mostly the roughness of the deposited film is strongly affected by the surface of the substrate. Therefore, authors should comment on the surface roughness of the optical fiber compared with SiO2 substrate.

In Fig. 1(b) and Fig. 2(b), letters are difficult to read. Please make them readable.

In Fig. 3(a), (1) please indicate the values of absorbance. Such values can give information about relative absorbance in the wavelength range. (2) Absorbance of ITO film is affected by the optical power. Please describe the power.

In Fig. 3(b), please indicate transmittance curves for different wavelengths to give wavelength dependencies.

Author Response

Reviewer 1:

This paper demonstrates a mode-locked operations in an fiber laser by using ITO SA. Compared with previously reported ITO SAs such as nanoparticles, nanoparticles in PVA or PMMA, they achieved uniform ITO film, which can be expected to endure large optical power and low insertion loss. Authors reported single-, dual-, triple-wavelength multipulses. The idea of ITO film SA is interesting, however, the detailed comparisons with previous researches and more analysis should be revised for the publication.

Comment 1Authors emphasized that ITO film can have excellent thermal stability, low insertion loss, and large scale. However, in the paper, they did not reported the threshold of optical power, insertion loss of their ITO films. Authors should discuss such values and compare them with those values of the previously reported ITO SAs.

Response to Reviewer comment: Thank the reviewer for the valuable comments. We have realized that more comparison with the previously work should be added in our text. Thence, we state several major aspects in our revised text, which motivate us to utilize ITO film. Firstly, the insertion loss was tested to be 1.4 dB. The values of the previously reported ITO SAs were not produced. Compared to the other PVA-type SAs, the insertion loss of the ITO film was significantly lower. For example, in Ref. [45, 57-59], the insertion losses were 5.5, 2.5, 3.5, and 6.1 dB, respectively. Secondly, the damage threshold of ITO film SA was much higher than the PVA-SA. Unfortunately, the damage threshold has not been tested in our experiment because of the limitation of our experimental conditions. In addition, the damage thresholds of other types of SA were not reported, so it is difficult to make a numerical comparison. To highlight the advantage of high damage threshold, some description had been added into the revised manuscript: As is known, the damage threshold of PVA was very low, so the ITO film sputtering on the fiber head has a higher damage threshold than the ITO/PVA SAs.

The corresponding modification has been made:

Please refer to the corresponding modification in revised manuscript:

Page 2, line 80-84.

Page 5, line 153-160.

Comment 2In Fig.1, in order to estimate the roughness of ITO film, authors used SiO2 substrate instead of optical fiber. However, mostly the roughness of the deposited film is strongly affected by the surface of the substrate. Therefore, authors should comment on the surface roughness of the optical fiber compared with SiO2 substrate.

Response to Reviewer comment: Thank the reviewer for the careful comments. It is our negligence that the explanation did not be placed in the text. We have added this explanation to the revised manuscript: The end face of the fiber core we used in this experiment was made of SiO2, so the parameters (including the thickness, the roughness and so on) of the ITO film deposited on the SiO2 substrate should be the same as the ITO film deposited on the fiber core.

The corresponding modification has been made:

Please refer to the corresponding modification in revised manuscript:

Page 3, line 104-110.

Comment 3In Fig. 1(b) and Fig. 2(b), letters are difficult to read. Please make them readable.

Response to Reviewer comment: Thank the reviewer for the valuable comments. It is our negligence that letters in these two figures are difficult to read. To make them easier to read, we redrew these two figures. It is worth mentioning that we have retested the EDS spectrum in order to get the raw data to draw a clearer figure. Therefore, the atomic ratio of In and Sn are slightly different than the original.

The corresponding modification has been made:

Fig 1(b) was replaced with

Please refer to Page 4 in the revised manuscript.

Fig 2(b) was replaced with

Please refer to Page 5 in the revised manuscript.

Please refer to more details in the revised manuscript

Page 4, line 127.

Comment 4In Fig. 3(a), (1) please indicate the values of absorbance. Such values can give information about relative absorbance in the wavelength range. (2) Absorbance of ITO film is affected by the optical power. Please describe the power.

Response to Reviewer comment: Thank the reviewer for the careful and valuable comments. (1) To report the absorbance in the wavelength range more clearly, the values of absorbance are very necessary. At 1558 nm, the absorbance was measured as 4.8 %. We’ve added the absorbance values and some descriptions into Fig. 3(a) and the text, respectively. (2) The absorbance of the ITO film was measured by employing an UV/vis/NIR spectrophotometer (Hitachi U-4100). The detection light sources are Xenon lamp (for UV region) and Tungsten halogen lamp (for vis/NIR region), and the power is at mW-level after passing through the spectrometer. However, due to the lack of instructions, we did not get the value of the specific power of the probe light. In order to reach the reviewer's requirements to the maximum extent within our capabilities, we have added the model number of the test instrument and the approximate value of the optical power into our revised text.

The corresponding modification has been made:

Fig 3(a) was replaced with

Please refer to Page 6 in the revised manuscript.

Please refer to more details in the revised manuscript

Page 5   Line 136-141.

Comment 5In Fig. 3(b), please indicate transmittance curves for different wavelengths to give wavelength dependencies.

Response to Reviewer comment: Thank the reviewer for the valuable comments. If the wavelength dependencies are added into the text, the characterization of ITO film will be more complete. However, since testing the nonlinear transmittance of different wavelength requires expensive laboratory equipment, our experimental conditions cannot meet these conditions. Therefore, we have no conditions to provide the transmittance curves for different wavelengths. In my opinion, it is more appropriate to test only the transmission curve of 1560 nm in this paper. Because the mode-locked Er-doped fiber laser was operated at the wavelength of ~ 1558.5 nm in this experiment. Therefore, the nonlinear transmittance (to characterize the saturable absorption at the wavelength where the fiber laser worked) was investigated at the wavelength of 1560 nm. Thank again for the valuable comments.

Reviewer 2 Report

In the manuscript submitted to the Nanomaterials the authors describe experimental results of saturation absorption (SA) studies based on the film-type ITO done by magnetron sputtering technique. The film-type SA exhibits an excellent thermal stability and low insertion losses. It is shown that this kind of films exhibit excellent nonlinear absorption characteristics, which can promote the use of ITO film for ultrafast photonics. In the manuscript we can find all necessary information including comprehensive introduction about current state of the art, description of materials and methods and finally results together with satisfactory discussion. The manuscript can be accepted in a present form

Author Response

Reviewer 2:

In the manuscript submitted to the Nanomaterials the authors describe experimental results of saturation absorption (SA) studies based on the film-type ITO done by magnetron sputtering technique. The film-type SA exhibits an excellent thermal stability and low insertion losses. It is shown that this kind of films exhibit excellent nonlinear absorption characteristics, which can promote the use of ITO film for ultrafast photonics. In the manuscript we can find all necessary information including comprehensive introduction about current state of the art, description of materials and methods and finally results together with satisfactory discussion. The manuscript can be accepted in a present form

Response to Reviewer: Thank the reviewer very much for the review and appreciation for our work.

Reviewer 3 Report

An interesting work that, in my opinion, deserves publication after some minor revisions. All the topics presented in the manuscript are not new, but the conjugation of them, and the quality of the ITO film produced, allowed a real nice result.

End of page 2 and beginning of page 3 – There is a good characterization of the ITO film preparation and characterization, but I couldn’t find the thickness of the film deposited in the fiber.

In section “2.3. Mode-Locked Fiber Laser” The Er-doped fiber is short (24 cm), so it should have a high Er concentration. I think that this parameter is more important than the dispersion for the laser. Please indicate it.

In figure 5c a modulation of the peak power at about 250 MHz can be seen. Is this a artifact, or a real effect? In either case it should be stated in the text.

In figure 6 we can see that the dual wavelength laser can be obtained at the same pump power than the single wavelength (287 mW). It seems that we can switch between them only by adjusting polarization controllers. Is this true? Is the adjusting of the PCs easy (reproducible) and stable (for how long can the laser work in one of the states without readjusting)?

The three wavelength laser is said to be obtained “at a higher pump power”. What is this value? Is it in the range of the two wavelength laser?

More than 40% self-citations! This number is too high to me. It appears that the authors are the only group working in mode-locked fiber lasers and film saturable absorbers. This percentage must be reduced bringing a more balanced view of the world work on the area.

I suppose that figures 5a, 6a, 7a and 8a where obtained with an OSA, and that the units should be dBm instead of dB

English usage should be looked after. I list below a few examples that I’ve came across:

Page 1, line 32 – “Saturable absorbers are the important”

Page 1, line 42 – “because the outstanding saturable”

Page 2, line 72, 73, 91 – “Besides” is not the best word, in my opinion.

Page 3, line 100 – “covered by a smooth ITO film with a large scale so that the light could be transmitted”

Page 3, line 103 – “The roughness of ITO film was obtained as 0.73”, missing units

Page 3, line 112 – missing definition of EDS

Page 3, line 115 – “This may be because a few Tin atomic were not”

Page 4, line 126 – “wavelength of 1560 nm and a reputation rate of 33.6 MHz.”

Page 4, line 127 – “The dependence of the transmission and the pump power”

Page 7, line 204 – “that the fundamental reputation rate remained 1.96 MHz.”

Page 7, line 220 – “data curation,”

Author Response

Reviewer 3: An interesting work that, in my opinion, deserves publication after some minor revisions. All the topics presented in the manuscript are not new, but the conjugation of them, and the quality of the ITO film produced, allowed a real nice result.

Comment 1End of page 2 and beginning of page 3 – There is a good characterization of the ITO film preparation and characterization, but I couldn’t find the thickness of the film deposited in the fiber.

Response to Reviewer comment: Thank the reviewer for the careful and valuable comments. It is our negligence that the thickness of the ITO film was not tested in our experiment. It is difficult to characterize the ITO film on the fiber directly, so the thickness of the ITO film deposited on the SiO2 substrate was investigated using AFM. The end face of the fiber core we used in this experiment was made of SiO2, so the parameters (including the thickness, the roughness and so on) of the ITO film deposited on the SiO2 substrate should be the same as the ITO film deposited on the fiber core. By analyze the AFM image, the thickness was obtained as 60 nm.

The corresponding modification has been made:

Fig 1(c) was added:                                       

Please refer to Page 4 in the revised manuscript.

Fig 1(d) was added:

Please refer to Page 4 in the revised manuscript.

Please refer to more details in the revised manuscript

Page 3   Line 104-115.

Page 4   Line 119-120.

Comment 2In section “2.3. Mode-Locked Fiber Laser” The Er-doped fiber is short (24 cm), so it should have a high Er concentration. I think that this parameter is more important than the dispersion for the laser. Please indicate it.

Response to Reviewer comment: Thank the reviewer for the valuable comments. As the reviewer said, the Er-doped fiber we used in this experiment was Er-110, which does have a high Er concentration. The peak core absorption, and the concentration of the Er3+ ion were 110 dB/m and 4350 ppm, respectively. These parameters were added into the revised manuscript.

The corresponding modification has been made:

Please refer to the corresponding modification in revised manuscript:

Page 6, line 164-166.

Comment 3In figure 5c a modulation of the peak power at about 250 MHz can be seen. Is this a artifact, or a real effect? In either case it should be stated in the text.

Response to Reviewer comment: Thank the reviewer for the careful and valuable comments. The amplitude envelope modulation was experimentally recorded as shown in Fig. 5(c). The digital oscilloscope used in our experiment is DPO4054. The bandwidth of the oscilloscope is 1 GHz and the sampling rate is 5GS/s. In addition, the oscilloscope signal is a digital signal. Therefore, the modulation of the peak power at about 250 MHz is measured realistically. This phenomenon was due to the power instability caused by the jitter of the laser and the phonon vibration of the SA material. We have added this explanation to the revised manuscript

The corresponding modification has been made:

Please refer to the corresponding modification in revised manuscript:

Page 9   Line 207-209.

Comment 4In figure 6 we can see that the dual wavelength laser can be obtained at the same pump power than the single wavelength (287 mW). It seems that we can switch between them only by adjusting polarization controllers. Is this true? Is the adjusting of the PCs easy (reproducible) and stable (for how long can the laser work in one of the states without readjusting)?

Response to Reviewer comment: Thank the reviewer for the careful comments. The dual-wavelength mode-locked operation can be realized at the same pump power than the single wavelength. By carefully adjusting the PCs, the polarization state in the cavity can be changed. Therefore, one mode-locked state maybe switches to another mode-locked state. This operation was reproducible and all the mode-locked state were highly stable. (We did not specifically test the stability of our mode-locked fiber laser for a long time, but at least, after all the characteristics had been recorded, the mode-locked states were still stable.)

Comment 5The three wavelength laser is said to be obtained “at a higher pump power”. What is this value? Is it in the range of the two wavelength laser?

Response to Reviewer comment: Thank the reviewer for the careful and valuable comments. It is our negligence that the value of the threshold was not mentioned in the text. The “higher pump power” that realized the triple-wavelength mode-locked state was about 420 mW, out of the range of the dual-wavelength mode-locked state. This value was added into the revised manuscript.

The corresponding modification has been made:

Please refer to the corresponding modification in revised manuscript:

Page 3, line 236.

Comment 6More than 40% self-citations! This number is too high to me. It appears that the authors are the only group working in mode-locked fiber lasers and film saturable absorbers. This percentage must be reduced bringing a more balanced view of the world work on the area.

Response to Reviewer comment: Thank the reviewer for the valuable comments. We have realized that the percentage of self-citations was too high. In the section of references, we have moved some our articles and added some works of other groups. The self-citations are 9/58 (16%) now.

The corresponding modification has been made:

Please refer to page 4, the “References” section in the revised manuscript.

Comment 7I suppose that figures 5a, 6a, 7a and 8a where obtained with an OSA, and that the units should be dBm instead of dB.

Response to Reviewer comment: Thank the reviewer for the careful and valuable comments. The unit here should really be dBm. We have replaced the units from dB to dBm in the revised manuscript.

The corresponding modification has been made:

Fig 5(a) was replaced with

Please refer to Page 8 in the revised manuscript.

Fig 6(a) was replaced with

Please refer to Page 10 in the revised manuscript.

Fig 7(a) was replaced with

Please refer to Page 11 in the revised manuscript.

Fig 8(a) was replaced with

Please refer to Page 11 in the revised manuscript.

Comment 8: English usage should be looked after. I list below a few examples that I’ve came across:

Page 1, line 32 – “Saturable absorbers are the important”

Page 1, line 42 – “because the outstanding saturable”

Page 2, line 72, 73, 91 – “Besides” is not the best word, in my opinion.

Page 3, line 100 – “covered by a smooth ITO film with a large scale so that the light could be transmitted”

Page 3, line 103 – “The roughness of ITO film was obtained as 0.73”, missing units

Page 3, line 112 – missing definition of EDS

Page 3, line 115 – “This may be because a few Tin atomic were not”

Page 4, line 126 – “wavelength of 1560 nm and a reputation rate of 33.6 MHz.”

Page 4, line 127 – “The dependence of the transmission and the pump power”

Page 7, line 204 – “that the fundamental reputation rate remained 1.96 MHz.”

Page 7, line 220 – “data curation,”

Response to Reviewer comment: Thank the reviewer for the careful and valuable comments. We checked the English usage and made some changes in the revised manuscript. But the usage of “data curation” is a requirement of the magazine. Therefore, we have not made corresponding changes. Thank again for the valuable comments.

The corresponding modification has been made:

Please refer to the corresponding modification in revised manuscript:

Page 1, line 32-33.

Page 1, line 41.

Page 2, line 72, 73, 94.

Page 3, line 102-103.

Page 4, line 125.

Page 4, line 129.

Page 5, line 146-148.

Page 9, line 220.

Page 11, line 240.

Round  2

Reviewer 1 Report

Authors revised all the issues raised by reviewer carefully. Now I believe that this paper can be published in Nanomaterials as it is.

Reviewer 3 Report

I'm satisfied with the revision of the manuscript.